# Matrix product state techniques for two-dimensional systems at finite temperature

Benedikt Bruognolo[1,2], Zhenyue Zhu[3], Steven R. White[3], E. Miles Stoudenmire[3],

**1** Physics Department, Arnold Sommerfeld Center for Theoretical Physics and Center for NanoScience, Ludwig-Maximilians-Universität München, 80333 München, Germany
**2** Max-Planck-Institut für Quantenoptik, Hans-Kopfermann-Str. 1, 85748 Garching, Germany
**3** Department of Physics and Astronomy, University of California, Irvine, CA 92697, USA

August 24, 2017

## Abstract

**The density matrix renormalization group is one of the most powerful numerical methods for computing ground-state properties of two-dimensional (2D) quantum lattice systems. Here we show its finite-temperature extensions are also viable for 2D, using the following strategy: At high temperatures, we combine density-matrix purification and numerical linked-cluster expansions to extract static observables directly in the thermodynamic limit. At low temperatures inaccessible to purification, we use the minimally entangled typical thermal state (METTS) algorithm on cylinders. We consider the triangular Heisenberg antiferromagnet as a first application, finding excellent agreement with other state of the art methods. In addition, we present a METTS-based approach that successfully extracts critical temperatures, and apply it to a frustrated lattice model. On a technical level, we compare two different schemes for performing imaginary-time evolution of 2D clusters, finding that a Suzuki-Trotter decomposition with swap gates is currently the most accurate and efficient.**

# 1   Introduction

Two-dimensional strongly correlated electron systems are a major frontier of condensed matter physics. Even after decades of intense investigation many questions remain about the behavior of paradigmatic two-dimensional (2D) systems such as the kagome Heisenberg antiferromagnet [1, 2, 3, 4, 5] and the Hubbard model [6, 7]. Controlled and accurate numerical techniques are central for making progress. Recent advances in numerics have, for example, led to a consensus regarding the existence of stripe order in the underdoped Hubbard model [7].

Methods to access 2D finite-temperature physics are a key area of numerical development. Temperature-dependent properties can signal phase transitions, provide evidence for subtle ground-state scenarios, and of course allow comparisons to real experimental conditions.

Two very useful methods for computing thermal properties of strongly correlated electrons are quantum Monte Carlo (QMC) and series expansion techniques. Both can access large system sizes and a wide temperature ranges, yet encounter serious limitations. High-temperature series expansions often fail to converge at or below a thermal phase transition. QMC suffers from the sign problem, preventing its use for most frustrated magnets and models of itinerant fermions (though there are notable exceptions in special cases [8, 9, 10, 11]).

The seriousness of these problems makes tensor networks a compelling alternative. For example, the density matrix renormalization group (DMRG) is an algorithm for obtaining ground states in the form of a matrix product state (MPS) that is not affected by the sign problem. Following the success of MPS techniques, other families of tensor networks have been proposed that are better suited for 2D systems [12] and critical phenomena [13].

Finite-temperature extensions of tensor network techniques include the purification, or ancilla, method [14, 15, 16] and the METTS (minimally entangled typical thermal state) algorithm [17, 18]. (There are also transfer-matrix approaches for finite temperature using MPS [19, 20], though we do not discuss them further.) The purification method directly computes the thermal density matrix using imaginary time evolution techniques—the approach works well for high temperatures but the cost to reach lower temperatures grows rapidly. To address the limitations of the purification method, the METTS algorithm blends imaginary time evolution with Monte Carlo sampling, enabling a less costly pure-state formalism.

In this work we demonstrate that with MPS techniques we can obtain state of the art results for spin models in two dimensions over a wide range of temperatures, using the pu-

rification method for higher temperatures and METTS for lower temperatures. Some of the systems we study have a significant sign problem, making them out of reach of most Monte Carlo techniques. Though the approaches we use are not affected by the sign problem, they are of practical interest only if they scale to low enough temperatures and large enough system sizes to accurately estimate 2D behavior, as we show they do. We also study systems that undergo finite-temperature phase transitions, another challenge for which our techniques turn out to be well suited.

We review the methods we use in Section 2 before discussing our results. Our first set of results are for the spin-$\frac{1}{2}$ Heisenberg model on the triangular lattice in Section 3.1. This model has a severe sign problem [21, 22] and we are able to obtain results competitive with the few other methods able to treat it. In Section 3.2 we study a ferromagnetic XXZ model on the square lattice and obtain very accurate results for the critical temperature, both for a case where QMC results are available and for a case with additional frustrating second-neighbor interactions.

For those wishing to reproduce or extend our results, we have made our codes publicly available at: https://github.com/emstoudenmire/finiteTMPS

## 2  Methods

In this section we briefly review the two finite-temperature MPS algorithms we use in our simulations. We discuss the technical challenges for these MPS-based methods in the context of 2D systems. A major drawback of using MPS for 2D systems is their limitation to only modest system sizes in the direction transverse to the MPS path. We describe one way to minimize finite-size effects by using a numerical linked-cluster expansion. A key component of using the purification and METTS finite-temperature techniques with MPS is evolving MPS in imaginary time. This is challenging to do efficiently for 2D systems, and we discuss how to deal with the effectively further-neighbor interactions which necessarily arise.

### 2.1  Finite-temperature MPS techniques

Equilibrium thermal properties of quantum systems are fully encoded in the thermal density matrix

$$\hat{\rho} = \frac{1}{Z} e^{-\beta \hat{H}} \tag{1}$$

where $Z$ is the thermal partition function and $\beta = 1/T$ the inverse temperature. Employing MPS techniques for imaginary time evolution, $\hat{\rho}$ can be directly computed by relying on the concept of purification. Or one can avoid purification and sample over a cleverly chosen set of pure states, the so-called METTS (minimally entangled typical thermal state). In this section we briefly review both of these MPS approaches.

*Density-matrix purification.–* Building on the ideas of purification, references [14, 15, 16] showed how to efficiently represent a thermal density matrix in an MPS framework. To this end, an auxiliary (or ancilla) space $A$ is introduced as a copy of the physical Hilbert space $P$. The auxiliary sites can be interpreted as a heat bath thermalizing the physical sites. Using the construction of an enlarged Hilbert space $\mathcal{H} = P \bigotimes A$, it is possible to construct the thermal

density matrix from a pure state $|\psi_T\rangle$ by tracing out the auxilary degrees of freedom:

$$\hat{\rho} = \mathrm{Tr}_A |\psi_T\rangle\langle\psi_T| \ . \tag{2}$$

Starting at infinite temperature ($\beta = 0$), the purified state can be easily constructed as a product state of maximally entangled pairs of one physical and one auxiliary site each. To make a measurement at some finite temperature $T = 1/\beta$, one evolves $|\psi_0\rangle$ in imaginary time up to $\beta/2$. (The Hamiltonian used for time evolution is just the one defining the original problem and acts as the identity on the ancillary space.) An arbitrary static observable $\hat{O}$ can then be evaluated by computing the overlap $\langle\psi_T|\hat{O}|\psi_T\rangle$, tracing out the auxiliary degrees of freedom.

The purification or ancilla approach works extremely well at high temperatures. But despite scaling polynomially with $\beta$, it becomes very costly in practice for temperatures well below the typical energy scales of the Hamiltonian.

*METTS.–* The minimally entangled typical thermal state algorithm (METTS) represents an alternative to purification [17, 18]. Instead of constructing the full density matrix, METTS generates a set of typical states $|\phi_{\boldsymbol\sigma}\rangle$ satisfying

$$e^{-\beta\hat{H}} = \sum_{\boldsymbol\sigma} P_{\boldsymbol\sigma} |\phi_{\boldsymbol\sigma}\rangle\langle\phi_{\boldsymbol\sigma}| \ , \tag{3}$$

with $P_{\boldsymbol\sigma}$ denoting the probability of measuring the system for a given $\beta$ in $|\phi_{\boldsymbol\sigma}\rangle$. Starting from any orthonormal basis $\{|\boldsymbol\sigma\rangle\}$, it can easily be shown that the following definition generates a set of states in agreement with typicality condition of Eq. (3),

$$|\phi_{\boldsymbol\sigma}\rangle = \frac{1}{\sqrt{P_{\boldsymbol\sigma}}} e^{-\beta\hat{H}/2}|\boldsymbol\sigma\rangle, \quad P_{\boldsymbol\sigma} = \langle\boldsymbol\sigma|e^{-\beta\hat{H}}|\boldsymbol\sigma\rangle. \tag{4}$$

Exploiting the freedom in the choice of the orthonormal basis $\{|\boldsymbol\sigma\rangle\}$, the METTS approach starts from a set of classical product states of the form $|\boldsymbol\sigma\rangle = |\sigma^1\rangle|\sigma^2\rangle...|\sigma^N\rangle$. These states represent the natural choice for a typical ensemble at infinite temperature, where the system should behave classically. Since their entanglement entropy starts out exactly zero and grows slowly during the imaginary-time evolution, they can typically be represented efficiently as MPS (hence the notion "minimally entangled").

A thermal measurement of an arbitrary static observable $O$ can be computed as

$$\langle\hat{O}\rangle_T = \frac{1}{Z} \sum_{\boldsymbol\sigma} P_{\boldsymbol\sigma} \langle\phi_{\boldsymbol\sigma}|\hat{O}|\phi_{\boldsymbol\sigma}\rangle \ . \tag{5}$$

Sampling the METTS ensemble randomly according to the probability distribution $P_{\boldsymbol\sigma}/Z$, this expectation value can be evaluated by taking the ensemble average of $\langle\phi_{\boldsymbol\sigma}|\hat{O}|\phi_{\boldsymbol\sigma}\rangle$. To construct a sample, a Markov chain of product states $|\boldsymbol\sigma\rangle$ is generated sequentially by the use of local measurements and then imaginary-time evolved to a specific temperature. We refer to Ref. [18] for details on the sampling algorithm and techniques to minimize autocorrelation effects.

*Applicability.–* Both methods are highly complimentary since they work best in opposite limits [23, 24]. Purification is highly accurate and efficient at high temperatures as it does not require any statistical sampling. However, the full thermal density matrix becomes much more costly to represent as a tensor network at low temperatures in comparison to the cost

of representing low-lying energy eigenstates or the pure states encountered in the METTS algorithm. This is specifically limiting for the 2D applications studied in this work, where the MPS representation of the density matrix quickly reaches the numerically feasible limits due to the additional entanglement in the system. Considering a typical 2D cluster of moderate size, we typically cannot reach temperatures significantly lower than the dominant energy scale (e.g. the spin coupling strength $J$) using purification.

In contrast, the METTS algorithm scales similarly to the ground state DMRG algorithm [18], allowing it to reach significantly lower temperatures. This feature particularly pays off in the context of 2D clusters, as it enables us to access relevant temperature regimes out of reach of purification. The METTS approach is less efficient than purification for higher temperatures, due to the extra sampling overhead. In Appendix B we present an example comparing the scaling of the MPS bond dimension with respect to temperature for both finite-temperature representations in the context of the triangular lattice Heisenberg model.

## 2.2 Finite-size restrictions

To understand the challenges of using MPS for 2D systems, it is helpful to recall the challenges of using the DMRG algorithm to compute a ground state MPS. In order to work with constant accuracy when using DMRG in 2D, the number of states kept in the MPS must be increased exponentially with respect to the width (transverse size) of the system. This limits the accessible system sizes and requires careful finite-size scaling. Nevertheless, DMRG has become a highly successful and competitive method for 2D systems mainly due to its flexibility, controllable accuracy, and access to the full many-body wavefunction.

Finite-temperature extensions of DMRG face the same system size restrictions. However, a non-zero temperature can often ease the finite-size limitations as correlation lengths are typically much shorter than at zero temperature. Therefore it often suffices to study narrow systems to extract information about the system's properties in the thermodynamic limit. Here we employ two different strategies to minimize finite-size effects in our finite-temperature MPS simulations:

*Purification plus NLCE.–* The first approach combines density-matrix purification with the numerical linked-cluster expansion (NLCE). NLCE is a powerful method to calculate an extensive observable $O$ of a lattice model directly in the thermodynamic limit, without having to perform calculations for the infinite system. Instead, NLCE employs measurements on finite-size clusters with open boundary conditions in both directions, while eliminating boundary and finite-size effects by a systematic resummation strategy [25, 26, 27]. In particular, we follow Ref. [28] and perform a modified NLCE procedure which takes only rectangular clusters into account and thus avoids the numerical bottleneck of computing cluster embeddings (see Appendix A for details). Using MPS based purification as finite-temperature cluster solver allows us to reach larger cluster sizes and higher expansion order than previously reported in the literature. Unfortunately, METTS is not a good candidate for a NLCE cluster solver because its errors are predominantly statistical, rather than systematic in nature (see Appendix A.3).

*METTS on cylinders.–* Our second scheme works analogously to most 2D DMRG calculations by taking open boundary conditions along the larger lattice direction (the length) and periodic boundary conditions along the smaller direction (the width). In this cylindrical setup one can first perform bulk-cylinder extrapolations based on the "subtraction trick" involving cylinders of various length [29]. The subtraction trick converges exponentially quickly once

the cylinder lengths exceed the correlation length. Then, if one reaches large enough cylinder widths, a second extrapolation can be performed to estimate properties in the thermodynamic limit. One advantage of using open boundaries along one direction is the possibility of adding boundary pinning fields which favor a particular symmetry-broken state in a parameter regime exhibiting spontaneous order. All of our METTS simulation are performed on cylinders.

## 2.3 Imaginary-time evolution of 2D clusters

One main ingredient of both finite-temperature algorithms introduced in Sec. 2.1 is an imaginary-time evolution that evolves the MPS starting from an infinite temperature state to a specific temperature. On a computational level, this represents the most expensive part of our calculations. Since we deal with various types of 2D clusters including systems with an enlarged Hilbert space in the purified setup, we include a discussion on the most important aspects. In particular, we focus on two schemes for MPS time evolution which are able to deal with long-ranged interactions of the Hamiltonian emerging from the mapping of the 2D cluster to a 1D chain. One approach is based on a combination of the Trotter decomposition and swap gates; the other uses a recently developed MPO approximation for the time evolution operator [30]. We conclude this section with an extended comparison of the two approaches in terms of accuracy and numerical efficiency.

*Suzuki-Trotter with swap gates.–* The simplest and most efficient setup to perform MPS time evolution for a 1D system with short-ranged interactions is the Suzuki-Trotter decomposition which splits the time-evolution operator into a product of local operators

$$e^{-\hat{H}\tau} \approx \prod_{\langle ij \rangle} e^{-\hat{h}_{ij}\tau}, \tag{6}$$

where $\hat{H} = \sum_{ij} \hat{h}_{ij}$. Suzuki-Trotter decompositions are in general very accurate approximations of the time-evolution operator, since they conserve important symmetries of the system dynamics [31]. The only error source originates from the non-commutativity of neighboring bond operators. The resulting so-called Trotter error can be controlled easily by choosing a small enough time step $\tau$ and using higher-order decompositions [31].

A Trotter-based time evolution is generally not applicable to systems with long-ranged interactions. However, a modified Trotter algorithm can be applied if interactions are restricted to two-body terms. To this end, one has to introduce the concept of swap gates. A swap gate switches the states of two identical sites and thus helps to modify the MPS in such a way that a non-local Trotter gate can be applied locally [18]. For each non-local bond operator $e^{-\hat{H}_{ij}\tau}$ the MPS is modified by a first set of swap gates so that site $i$ is moved to the position of site $j-1$. The bond operator can now be applied locally before a second set of swap gates moves site $i$ back to its original position.

This scheme conserves the accuracy of the Suzuki-Trotter decomposition and, at the same time, can handle two-body interactions of any range. Nevertheless, its efficiency is strongly range-dependent. For the typical example of a rectangular 2D cluster considered in the following, the number of swaps scales roughly quadratically with the width of the system $N_y$. Since each additional swap requires an additional singular value decomposition computation, the method can become inefficient for wide systems.

*MPO decomposition.–* An alternative strategy relies on matrix-product-operator (MPO) approximations of the evolution operator $e^{-\hat{H}\tau}$ that can naturally include long-ranged interaction terms. An MPO-based time evolution is especially favorable for systems with different

types of long-ranged interactions, such as exponentially decaying terms which cannot be captured nicely in terms of two-site gates but which are encoded efficiently in the MPO representation of the Hamiltonian [32]. Although such systems are not considered in this work, an MPO-based approach could conceivably have better efficiency than the Trotter and swap gate approach when working on large 2D clusters. Hence we benchmark our Trotter scheme against the recently developed MPO-based scheme of Ref. [30].

An appealing feature of the approach of Ref. [30] is the enhanced error control in comparison to established MPO approximations, such as a simple Euler step or its Runge-Kutta and Krylov extensions. The key insight of Ref. [30] is to improve the simple Euler step by a local version of the Runge-Kutta stepper

$$1 + \tau \sum_x \hat{H}_x \quad \rightarrow \quad \prod_x (1 + \tau \hat{H}_x). \tag{7}$$

Using this approximation, the error remains constant with system size. In contrast an Euler stepper would incur an error per site that diverges for large systems. In addition, the Ref. [30] approach gives a very compact MPO representation making it appealing in terms of efficiency and implementation. The actual time evolution is carried out by applying the MPO to a MPS using standard tools, such as the fitting approach [12]. Note that one can combine two complex time steps to further reduce the scaling of the error per step to $\mathcal{O}(\tau^3)$. This leads to a second-order decomposition of the evolution operator which we use in our tests below.

*Discussion.–* In the following we compare the two schemes introduced above with respect to accuracy and numerical efficiency. The accuracy of the imaginary-time evolution is impaired by two error sources. On the one hand, the approximative decomposition of the full time-evolution operator introduces a finite time-step error. On the other hand, the accuracy is affected by the dynamical truncation of the MPS during the time evolution, which we control by adapting the cutoff parameter $\epsilon$ that limits the maximum discarded weight when truncating the MPS using a singular value decomposition [33]. These two error sources are not fully independent. Whereas a smaller time step decreases the effects of the decomposition error, it might increase the influence of the truncation error since more individual truncation steps are required during the time evolution. The numerical costs strongly depends on the total number of time steps as well. Fewer steps typically reduce the total computational time. In addition, the required MPS bond dimension might differ depending on the evolution scheme and time step leading to a slightly different cost scaling.

In our analysis we focus on a spin-$\frac{1}{2}$ antiferromagnetic XY model with nearest-neighbor interaction on a square-lattice cluster of size $5 \times 5$ with open boundary conditions. This cluster represents a nontrivial system with long-ranged interaction, yet it is still small enough to generate a quasi-exact reference state required for a proper benchmark procedure. Starting with a Neel-state at $\beta = 0$, we evolve the system in imaginary time to $\beta = 4$ and track the performance of both evolution schemes in terms of accuracy and numerical efficiency. The accuracy is monitored by calculating the overlap with respect to a quasi-exact reference state [34] after each time step $\tau$ for a cutoff $\epsilon = 10^{-10}$. Moreover, we track the CPU time required for each time step and the bond dimension $m$ of the MPS during the evolution. We show the results in Fig. 1.

To our surprise, it turns out that the Trotter approach not only gives the more accurate results but also requires significantly less numerical resources. Studying the deviation from the exact state in Fig. 1(a), it becomes apparent that the Trotter approach is dominated by the truncation error. Reducing the time step $\tau$ by one order significantly decreases the accuracy

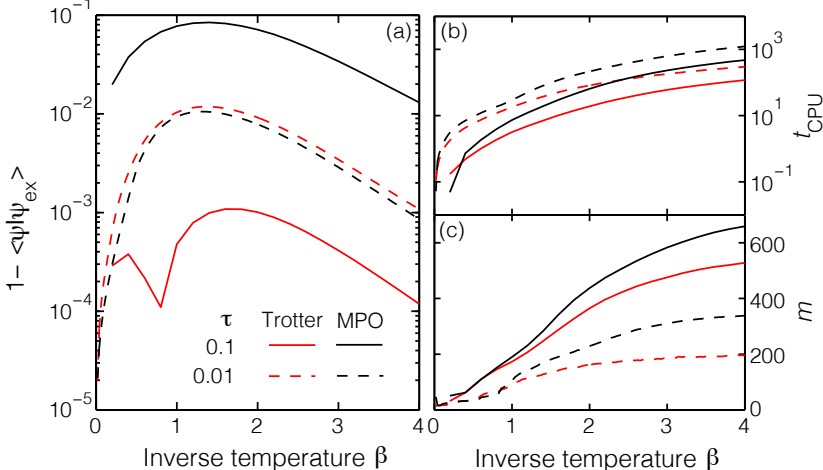

Figure 1: Comparison of MPO- and Trotter-based imaginary-time evolution for a spin-$\frac{1}{2}$ XY model on $5 \times 5$ square cluster with open boundary conditions.

due to the additional number of truncations. In contrast, the accuracy of MPO scheme strongly improves with a smaller time step indicating that the time step error represents the main error source. Nevertheless, the overall accuracy still remains roughly one order of magnitude larger than the Trotter approach with $\tau = 0.1$. At the same time, the MPO approach requires both significantly more CPU time and a larger bond dimension $m$, as shown in Fig. 1(b) and (c), rendering it less efficient than the Trotter scheme.

We conclude that the combination of a Trotter decomposition with swap gates currently represents the best choice for the purposes of this work, namely carrying out imaginary-time evolution for systems with long-ranged two-body interaction terms. At least the simple variant of the MPO decomposition of Ref. [30] employed here cannot meet the standards of the Trotter approach in terms of accuracy or efficiency. It may be that the MPO approximation misses some relevant contributions to the time evolution operator $e^{-\tau \hat{H}}$ that are included in the Trotter approximation. This could account for why the MPO approach requires smaller time steps to minimize the decomposition error. The more complex MPO variant of Ref. [30] might improve this behavior. However, due to its model dependent implementation we here refrained from testing it also. Moreover, the implementation of the second-order MPO decomposition requires complex numbers for imaginary-time steps. This increases the computational complexity in comparison to Trotter. We note that this is special to finite-temperature calculations. In the context of real-time evolution all time-evolution approaches require complex numbers, hence, the efficiency of the MPO approach might improve here.

Recent years have seen interesting developments regarding time evolution algorithms in systems with long ranged interactions. In addition to the two approaches discussed here, other suitable techniques include the time-dependent variational principle [35] or a recently introduced series-expansion thermal tensor network [36]. At the moment, these approaches coexist independently and, due to the inherent technical complexity, a practitioner typically picks the one most suitable to his problem and implementation framework. It remains an open question whether there exists a "best practice" approach amongst these schemes. Hence, a detailed benchmarking of these various time evolution techniques including different systems

and both real and imaginary time would be extremely helpful and is left for future work.

# 3   Results

Based on the MPS techniques introduced above, we present calculations for two frustrated lattice spin-$\frac{1}{2}$ systems in this section. First we focus on the antiferromagnetic Heisenberg model on the triangular lattice, illustrating that our MPS approach agrees with and extends results from other state of the art techniques to lower temperatures in the zero-field limit. In addition we also explore the system including a finite magnetic field. Moreover, we show how a METTS based approach can be used to detect a finite-temperature phase transition in a frustrated XXZ model on the square lattice. All calculations in this work are performed using the ITensor library [37]. To keep the discussion compact, we refer to Appendix B for further numerical details regarding the MPS parameters and setup.

## 3.1   Triangular lattice Heisenberg model

The antiferromagnetic spin-$\frac{1}{2}$ Heisenberg model (AFMH) on the triangular lattice, defined as

$$\hat{H} = J \sum_{\langle i,j \rangle} \hat{\mathbf{S}}_i \cdot \hat{\mathbf{S}}_j - h_z \sum_i \hat{S}_i^z \,, \tag{8}$$

represents a paradigmatic example of a frustrated lattice model (we assume $J = 1$ in what follows). It has been suggested as an effective description for several compounds such as $Ba_3CoSb_2O_9$ [38, 39], $Cs_2CuBr_4$ [40, 41] and, most recently, $Ba_8CoNb_6O_{24}$ [42, 43]. The combination of geometric frustration, reduced dimensionality and $S = 1/2$ degrees of freedom significantly enhances quantum fluctuations.

Without external magnetic field, $h_z = 0$, the ground state of the system exhibits coplanar, long-range magnetic order. The three spins in each triangle arrange themselves at $120°$ to one another in the same plane, forming a three-sublattice structure. For a finite magnetic field, the system features a broad magnetization plateau at $\frac{1}{3}$ of the total magnetization, where the spins order in a collinear, "up-up-down" configuration. Two coplanar phases surround the magnetization plateau transforming into a fully polarized state at very large fields.

At zero temperature, the model has been intensively studied by a number of methods, ranging from semi-classical approaches to extensive DMRG calculations pinning down the ground state phase diagram with high precision [45, 46, 47, 48, 49]. However, the finite-temperature regime still remains elusive since conventional QMC is impaired by the sign problem [21, 22]. In the zero-field limit, alternative approaches have proven very useful, such as high-temperature series expansions [50, 44, 51], conventional numerical cluster expansions [26], bold diagrammatic QMC [9], and Schwinger-Bosons [52]. Nevertheless, the limitation of all these methods in terms of accessible temperatures or system sizes leave room for improvement. In the following, we show that the combination of purification and NLCE, supported by additional METTS results on cylinders at lower temperatures, can be a competitive approach for determining the finite-temperature properties of the triangular AFHM.

*Zero-field limit.–* The ordered ground state of the triangular lattice AFHM at zero magnetic field breaks only the continuous SU(2) symmetry of the Hamiltonian, as elaborated above. In this case spontaneous symmetry at finite temperature is prohibited by the Mermin-Wagner theorem [53], so that no phase transition can be observed at finite temperatures.

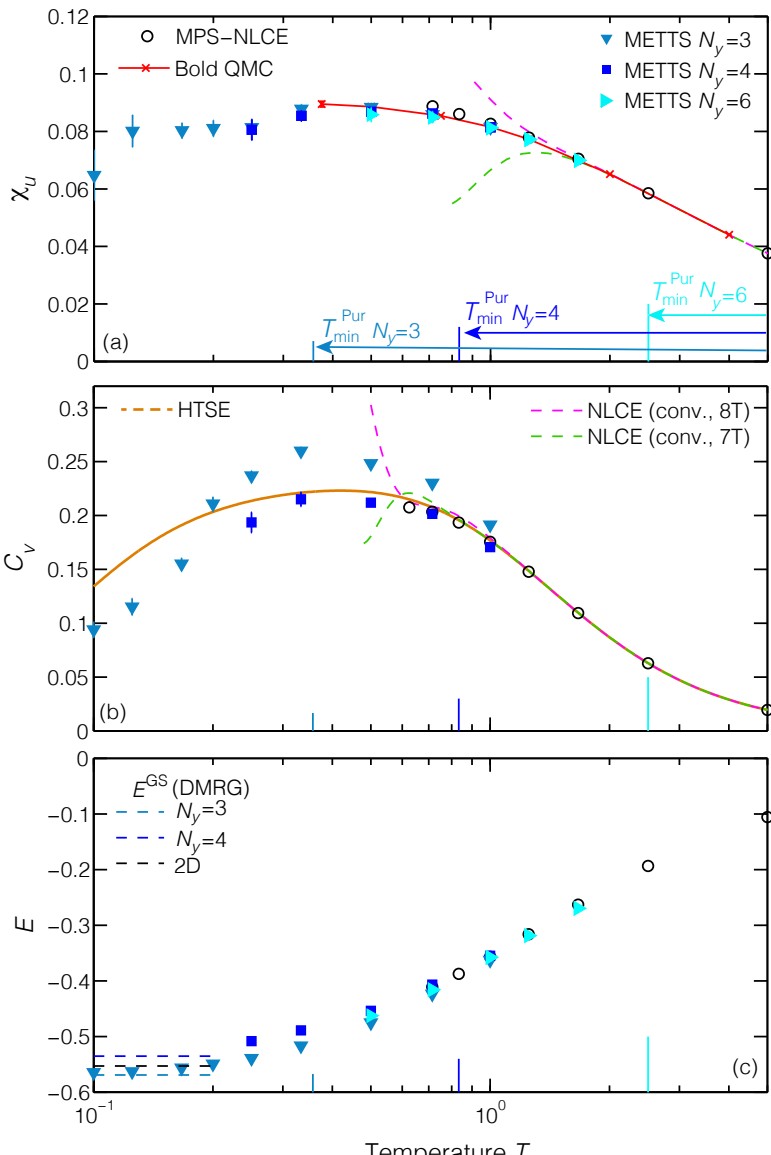

Figure 2: NLCE and METTS calculation of (a) uniform susceptibility $\chi_u$, (b) specific heat $C_v$, and (c) energy density $E$ (all per site) as a function of temperature for the triangular AFHM in the zero-field limit, $h_z = 0$. In comparison, we show results from high-temperature series expansion (HTSE) [44] for the specific heat, as well as bold diagrammatic QMC [9] and conventional NLCE [26] for the susceptibility. The colored ticks indicate the accessible temperature ranges when using density-matrix purification on width $N_y = 3, 4, 6$ cylinders while enforcing the same numerical accuracy as in the METTS calculation. This illustrates the importance of METTS for performing calculations at low temperatures (see Appendix B.1 for details).

Spin correlations remain short-ranged down to rather low temperatures (e.g. for $T = 0.25$ the correlation length is about two lattice spacings [50, 25]). This makes the model particularly suitable for our MPS techniques, which are primarily limited by system width. Our MPS calculations for this system employ a second-order Trotter decomposition with time step $\tau = 0.1$ and a truncation error cutoff $\epsilon = 10^{-10}$ for $N_y = 3, 4$, while for $N_y = 6$ we use a variable cutoff strategy, varying the cutoff from $\epsilon = 10^{-11}$ up to $\epsilon = 10^{-8}$ throughout different steps of the imaginary time evolution. The MPS-NLCE results are obtained on open clusters, whereas METTS is performed on long cylinders and finite-length effects are minimized via extrapolation. See Appendix B.1 for more details.

Our results for the zero-field case are presented in Fig. 2, where we study the finite-temperature properties of the triangular AFHM in terms of uniform susceptibility $\chi_u = (\langle \hat{\mathbf{S}}^2 \rangle_T - \langle \hat{\mathbf{S}} \rangle_T^2)/(3TN)$ [Fig. 2(a)], specific heat $C_v = (\langle \hat{H}^2 \rangle_T - \langle \hat{H} \rangle_T^2)/(T^2 N)$ [Fig. 2(b)], and energy density $E = \langle \hat{H} \rangle_T / N$ [Fig. 2(c)] per site, where $\hat{\mathbf{S}} = \sum_i \hat{\mathbf{S}}_i$. For the purpose of benchmarking, we compare our calculations with other state of the art techniques. This includes results from high-temperature series expansion (HTSE) [44] and conventional NLCE [26] in context of $C_v$, as well as bold diagrammatic QMC for $\chi_u$ [9].

Our MPS results show excellent agreement with the benchmark data, highlighting the complementarity of the two MPS-based strategies. First, we cover the high temperature regime shown in Fig. 3 down to about $T \sim 0.7$ with MPS-NLCE based on purification applied to cluster with a maximum size of $5 \times 5$ sites (black circles). We can observe that MPS-NLCE converges to significantly lower temperatures compared to conventional NLCE for quantities such as the susceptibility [Fig. 2(a)] since it can access larger cluster sizes. The resolution at high temperatures can easily be improved by using a smaller Trotter step, as illustrated in Fig. 3.

Lower temperatures are reached with METTS calculations on cylinders of width $N_y = 3$ (darker, downward triangular symbols), $N_y = 4$ (squares), and $N_y = 6$ (lighter, rightward triangles) shown in Fig. 2. The additional colored ticks along the temperature axis illustrate the minimum temperature accessible to purification on such cylinders with our resources, indicating that METTS algorithm is crucial for reaching the lowest temperatures shown. Cylinders of $N_y \leq 4$ turn out to be sufficient for estimating 2D properties down to rather low temperatures, depending somewhat on the specific property one is calculating. For $\chi_u$, we find very good agreement with bold diagrammatic QMC down to the lowest temperature data currently available, about $T = 0.375$. For even lower temperatures down to $T = 0.25$, the results for $N_y = 3$ and $N_y = 4$ cylinders continue to agree, indicating that our $N_y = 4$ results for $\chi_u$ in Fig. 2(a) have very small finite-size effects.

Finite-size effects are clearly more significant for our $C_v$ results Fig. 2(b). However, our $N_y = 4$ data is mostly in agreement with estimates of $C_v$ based on high-temperature series calculations [44, 26]. We also show the energy per site in Fig. 2(c) to get further information about finite-size effects and to demonstrate that the lowest temperatures reached bring the system close to its ground state.

*Finite magnetic field.–* The triangular AFHM features four distinct magnetically ordered states at low temperatures for various values of an applied magnetic field [49]. The most striking state is the collinear, "up-up-down" configuration forming an extended magnetization plateau for $1.3 < h_z < 2.1$ at $\frac{1}{3}$ of the total magnetization. This plateau state is surrounded by two different states with co-planar order. In the high field region, $h_z > 4.5$, the system is fully polarized in the direction of the applied field. As some of these ordered phases spontaneously

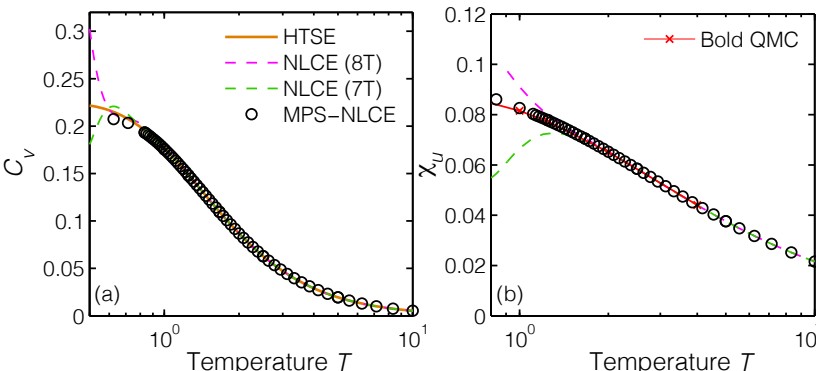

Figure 3: MPS-NLCE with smaller Trotter time step $\tau = 0.01$ illustrating excellent agreement with reference data at high temperatures.

break a discrete symmetry, they can persist for finite temperatures $T > 0$ and in such cases will be separated from the high-temperature paramagnetic phase by a finite-temperature phase transition.

Most theoretical studies of the triangular AFHM in an external magnetic field focus on the zero-temperature phases. The finite-temperature phase diagram has been explored by Monte-Carlo in the large-S limit [54] as well as in experimental AFHM in materials such as $Ba_3CoSb_2O_9$ [38] and $Cs_2CuBr_4$ [41].

Here we apply our MPS techniques to gather additional insight into the finite-temperature properties of the system. Unfortunately, we are not able to fully resolve the finite-temperature phase transition since, according to the experiments, the ordered phases should appear only for low temperatures $T < 0.25$. Our METTS sampling is numerically limited to temperature regimes of $T > 0.25$ on width $N_y = 4$ systems. However, an explicit study of $T_c$ might be in reach employing a recently introduced, more efficient METTS sampling, that allows symmetry conservation even in the presence of a magnetic field [55]. This is left for future work.

For now we present MPS results for finite $h_z$ in Fig. 4, which includes energy density $E$ [Fig. 4 (a)], magnetization $m_z$ [Fig. 4 (b)], specific heat $C_v$ [Fig. 4 (c)], and susceptibility $\chi_z = (\langle \hat{m}_z^2 \rangle_T - \langle \hat{m}_z \rangle_T^2)/(TN)$ [Fig. 4 (d)] per site. We focus on three different field strengths, each representing a point in either one of the two distinct co-planar phases ($h_z = 1, 3$) or in the plateau phase ($h_z = 2$). Again the high-temperature regime is covered by our MPS-NLCE using purification (solid lines) while METTS is used at lower temperatures (squares).

The presence of long-range ordered states at finite-temperature can lead to stronger spin-spin correlations at higher temperatures if the system undergoes a continuous transition. This can clearly be seen in context of $C_v$ and $\chi_z$ for which the NLCE breaks down (due to finite cluster size effects) at significantly higher temperatures in comparison to the zero-field case. This breakdown of MPS-NLCE is indicated by dashed lines in Fig. 4(c,d) showing the naive continuation of our MPS-NLCE procedure to lower temperatures for fixed maximum cluster size. It is unlikely that this is already a clear signature of the critical temperature $T_c$ since we expect the phase transition to appear at significantly lower temperatures [38]. Moreover, the effect is most pronounced for strong magnetic fields at $h_z = 3$, whereas $T_c$ should actually decrease compared to the plateau phase at $h_z = 2$.

Our $N_y = 4$ METTS results agree well with NLCE at high temperatures and can be pushed

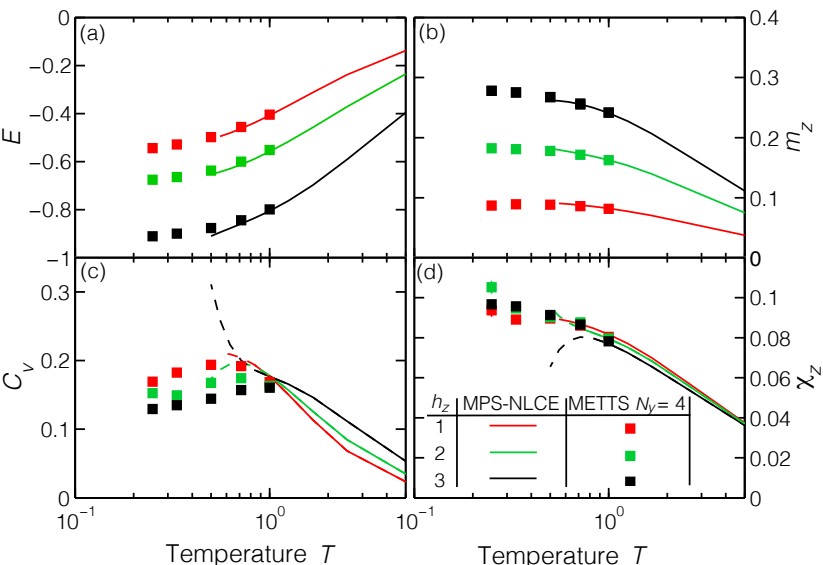

Figure 4: NLCE and METTS calculations of (a) energy density $E$, (b) magnetization $m_z$, (c) specific heat $C_v$, and (d) susceptibility $\chi_z$ (all per site) as a function of temperature for the triangular AFHM with finite magnetic field. The different field strengths include the two co-planar phases ($h_z = 1, 3$) and the plateau phase ($h_z = 2$).

to substantially lower temperatures than MPS-NLCE can reach for the same resources. An interesting feature appears in the METTS results for $h_z = 2$ where the magnetization already tends to saturate at $\frac{1}{3}$ of the total magnetization. This indicates that the system might be already ordered for the lowest temperature accessible, $T = 0.25$, in agreement with Ref. [38]. Nevertheless, we expect finite-width effects to be more pronounced in Fig. 4 than in the zero-field results shown in Fig. 2.

## 3.2 $J_1$-$J_2$ XXZ model on the square lattice

In this section we show next that METTS is capable of accurately detecting a finite-temperature phase transition in lattice models with and without frustration, despite being limited to finite-size cylinders. This is a very appealing feature since the sign problem, often present in frustrated or fermionic systems, severely limits Monte Carlo techniques. Other powerful techniques such as high-temperature series expansion typically fail above or at the critical point, such that one has to rely on perturbative or mean-field approaches to determine the critical properties in such systems. While such approaches can work adequately on a qualitative level, they introduce significant quantitative errors such as in the determination of $T_c$. Our MPS scheme, which is neither limited by the sign problem nor by strong quantum fluctuations and interactions, could offer a valuable alternative to extract the exact location of the critical point in such models, assuming one can reach large enough system sizes for a particular problem of interest.

As proof of principle for our MPS approach, we consider a spin-$\frac{1}{2}$ $J_1$-$J_2$ XXZ model on

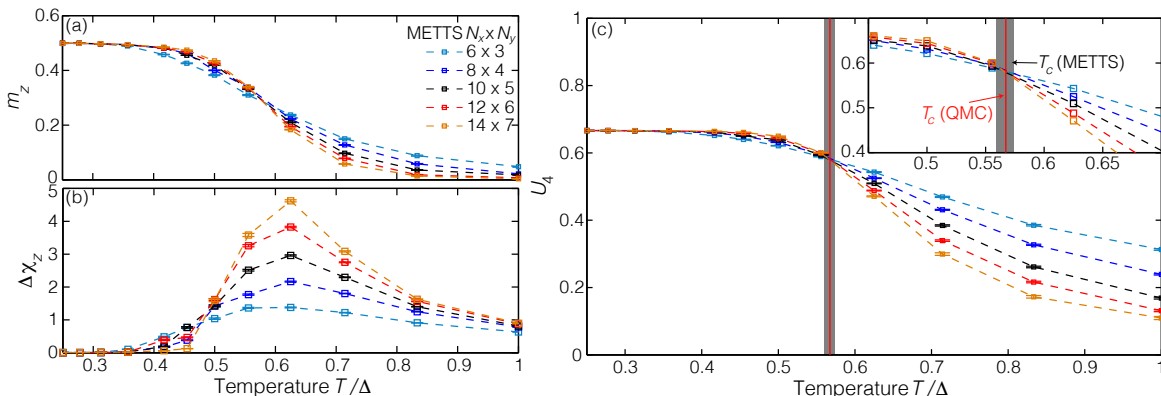

Figure 5: $T_c$ estimate for non-frustrated spin-$\frac{1}{2}$ XXZ model on square lattice using METTS ($\Delta = 5$, $J_2/|J_1| = 0$). (a) Magnetization $m_z$, (b) susceptibility $\chi_z$, and (c) forth-order Binder cumulant $U_4$ as a function of temperature for different system sizes. Inset in (c) shows universal crossing point of Binder cumulants indicating the location of critical temperature $T_c/\Delta = 0.56 \pm 0.01$ (vertical shading) in excellent agreement with quantum Monte-Carlo calculation (vertical red line) [56].

the square lattice,

$$
\begin{aligned}
\hat{H} &= J_1 \sum_{\langle i,j \rangle} \left( \hat{S}_i^x \hat{S}_j^x + \hat{S}_i^y \hat{S}_j^y + \Delta \hat{S}_i^z \hat{S}_j^z \right) \\
&\quad + J_2 \sum_{\langle\langle i,j \rangle\rangle} \left( \hat{S}_i^x \hat{S}_j^x + \hat{S}_i^y \hat{S}_j^y + \Delta \hat{S}_i^z \hat{S}_j^z \right),
\end{aligned}
\tag{9}
$$

with ferromagnetic nearest-neighbor (NN) coupling $J_1 = -1$, antiferromagnetic next-nearest-neighbor (NNN) coupling $J_2 > 0$ and exchange anisotropy $\Delta > 1$ chosen in the following.

*Non-frustrated model.*– We first consider the non-frustrated nearest-neighbor XXZ model as a benchmark ($J_2 = 0$). In this case QMC is fully applicable and both ground-state and finite-temperature phase diagrams are well established [57, 58, 59]. Here we focus on the easy-axis regime $\Delta > 1$ where the spins in the system order ferromagnetically along the $z$-axis below some critical transition temperature $T_c$. The order parameter is given by the total magnetization per site $m_z = \langle \hat{S}^z \rangle / N$, with $\hat{S}^z = \sum_i \hat{S}_i^z$. This transition corresponds to a spontaneous breaking of the $Z_2$ symmetry of the Hamiltonian and belongs to the same universality class as the phase transition of the 2D Ising model.

In the following, we choose $\Delta = 5$ and assess whether we can detect the critical point with reasonable accuracy using METTS calculations on cylinders (see Appendix B.2 for numerical details). To this end, we employ the concept of a Binder cumulants [60], a method very commonly used in Monte Carlo studies to pin down the precise value of a critical point. Tensor network techniques have made use of Binder cumulants only very occasionally, and then only in the context of quantum phase transitions in 1D and quasi-1D systems [61, 62]. Here we show that the applicability can be straight-forwardly extended to thermal phase transitions in 2D models as well.

The Binder cumulant is a particularly useful quantity to study the critical point in systems with a known order parameter. For a system with $Z_2$ order parameter, such as (9), this

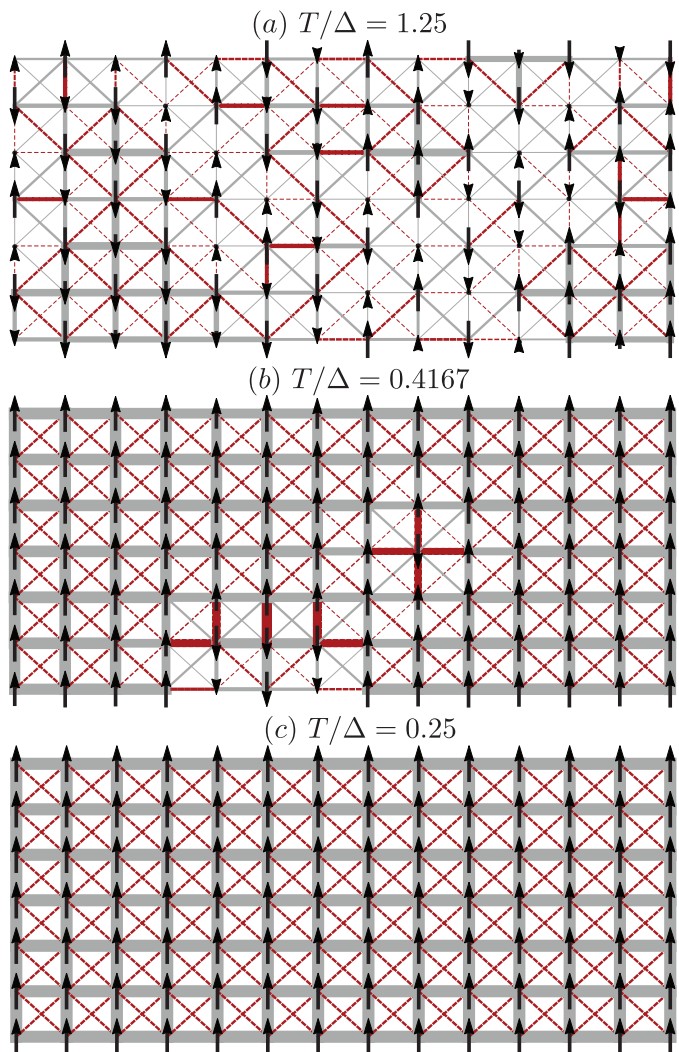

Figure 6: Snapshot of individual METTS in different phases of frustrated XXZ model ($\Delta = 5$, $J_2/|J_1| = 0.2$) on width $N_y = 7$ cylinder (a) in paramagnetic phase, (b) close to the phase transition, and (c) in ferromagnetically-ordered phase. The size of the arrows represent local measurements of $\langle \hat{S}^z \rangle$, and the widths of lines proportional to a bond measurement $J_{1/2}\langle(\hat{S}_i^x \hat{S}_j^x + \hat{S}_i^y \hat{S}_j^y + \Delta \hat{S}_i^z \hat{S}_j^z)\rangle$. Gray lines indicate a negative value of the bond measurement, while red lines correspond to positive bond measurements.

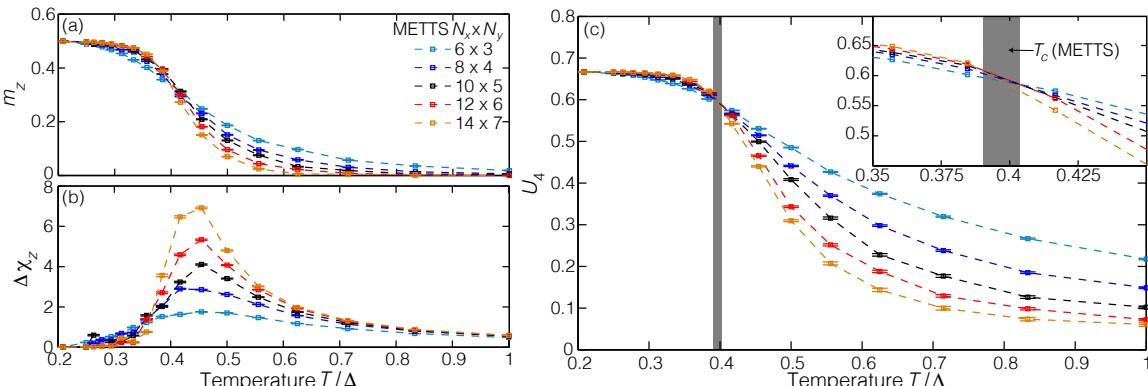

Figure 7: Same METTS-based $T_c$ calculation as in Fig. 5 but for frustrated spin-$\frac{1}{2}$ XXZ model with next-nearest-neighbor interactions on square lattice ($\Delta = 5$, $J_2/|J_1| = 0.2$). Analogously to the non-frustrated case, $U_4$ enables us to precisely extract the value of $T_c/\Delta \approx 0.39 \pm 0.01$.

modified forth order moment is defined as

$$U_4 = 1 - \frac{\langle (\hat{S}^z)^4 \rangle}{3 \langle (\hat{S}^z)^2 \rangle^2} \,. \tag{10}$$

Note that the Binder cumulant can be obtained for other symmetry-broken orders as well, but the prefactors are typically different. The special feature of the Binder cumulant is its very distinct dependence on system size in the different phases as well as close to the phase transition. For $T > T_c$ in the disordered phase $U_4$ approaches zero with increasing system size, whereas it converges to a constant value $U_4 = \frac{2}{3}$ in the ordered phase for $T < T_c$. Close to the critical point the Binder cumulant is only weakly dependent on system size. Thus the curves of $U_4$ as a function of temperature, plotted for several system sizes, should all intersect at the critical temperature $T = T_c$. Due to the universality at the critical point, one can typically determine $T_c$ very accurately on small systems without having to perform complex extrapolations to the thermodynamic limit. This makes the concept of Binder cumulants particularly appealing for MPS applications which are limited to modest system widths in 2D.

Fig. 5 shows our METTS calculations on cylinder with varying system sizes with constant aspect ratio (ratio of length to width is chosen 2:1). We set time step $\tau = 0.1$ and cutoff $\epsilon = 10^{-10}$ in all calculations and, in addition, apply pinning fields on the open boundaries of the system (see Appendix B.2 for more details).

While the precise critical temperature cannot be easily read off from the order parameter behavior [Fig. 5(a)], the maximum of susceptibility [Fig. 5(b)] gives already a first rough estimate of $T_c/\Delta \sim 0.6$. Studying the crossings of Binder cumulant obtained from calculations on different system sizes [Fig. 5(c) and inset], we obtain a much more precise estimate of $T_c/\Delta = 0.56 \pm 0.01$ which is indicated by the shaded region. This interval includes all line-segment crossings of $U_4$ and serves as a conservative error bar estimate for $T_c$. We are pleased to find this result in excellent agreement with QMC calculations [56], indicated by the vertical red line.

*Frustrated model.–* One of the attractive features of numerical approach is its straightforward adaptability to frustrated systems. As long as the states we encounter during imaginary-

time evolution have sufficiently low entanglement, we can straightforwardly treat the model with a finite antiferromagnetic NNN coupling $J_2 > 0$ that adds frustration. In this way, our METTS approach is a rare method that neither suffers from the sign problem nor contains a perturbative or mean-field ansatz and could become very helpful for studying finite-temperature transitions in frustrated systems.

In the following, we consider the Hamiltonian in Eq. (9) with $\Delta = 5$ (as before) and $J_2 = 0.2|J_1|$, which is small enough to preserve the ferromagnetic order at low temperatures. For the system with the additional $J_2$ interaction, we illustrate the physical properties of the system in the disordered phase, in the ordered phase, and close to the critical point in Fig. 6, showing individual METTS on a width $N_y = 7$ cylinder as "snapshots" of the different temperature regimes. In the paramagnetic phase at high temperatures, $T > T_c$, a typical METTS such as Fig. 6(a) is dominated by strong thermal fluctuations. As expected in a paramagnetic phase, we observe no indication of magnetic ordering, not even on the level of small clusters. Close to the critical temperature the METTS shown in Fig. 6(b) is already ferromagnetically ordered in most parts of the system. However, thermal fluctuations are still strong enough to flip individual spins or even small clusters, weakening the ferromagnetic order. For temperatures below the critical point, $T < T_c$, the ferromagnetic order is very strong, as illustrated by the METTS in Fig. 6(c).

Studying the independent METTS samples for different temperatures, one can notice another interesting yet intuitive effect. The temperature values requiring the most computational effort are in the paramagnetic phase very close to the phase transition. There the METTS are subject to thermal fluctuations which become long-ranged in the vicinity of the critical point, leading to large fluctuations in the local properties of the METTS and large sample variance. At lower temperatures the system orders ferromagnetically, breaking the $Z_2$ spin symmetry. The METTS in the ferromagnetic phase have relatively smaller fluctuations and, when collapsed, result in largely similar product states with most spins aligned. Empirically, we find that for these lower temperatures the sample variance is both smaller and the METTS are typically less entangled than for ensembles sampled just above the critical temperature. Therefore much less computational effort is required to get good accuracy.

Compared to the purification method, where the effort needed to reach lower temperatures is always strictly greater than for higher temperatures, in the above scenario of applying METTS within a low-temperature ordered phase we find that the METTS algorithm bypasses much of the numerical difficulties associated with the thermal phase transition. By this we mean that the algorithm adapts to the simpler low-temperature physics of an ordered phase, and does not require one to first deal with higher-temperature properties as a necessary precursor to obtaining low-temperature properties.

Turning again to the detection of $T_c$, we use the same procedure as for the non-frustrated case and the corresponding results are shown in Fig. 7. While the temperature dependence of the order parameter [Fig. 7(a)] and the susceptibility [Fig. 7(b)] again only allow rough estimates of the exact value of the critical temperature, the Binder cumulant [Fig. 7(c)] enables us to determine $T_c$ much more precisely. Plotted as a function of temperature, the Binder cumulants approximately cross at an universal point indicating $T_c/\Delta \approx 0.39 \pm 0.01$ [see inset of Fig. 7(c)]. Again, the shaded region contains all line-segment crossings of $U_4$ and serves as a conservative error bar estimate.

Despite the small value of $J_2$, we note that $T_c$ decreases by almost 30% in comparison to the non-frustrated model since the $J_2$ couplings move the system closer to a regime where the ground state of the system is described by a stripy antiferromagnet. The ferromagnetic order

is expected to vanish for $J_2/|J_1| \approx 0.4$ in the isotropic model ($\Delta = 1$) [63]. The suppression of the thermal phase transition to lower temperatures is congruent with previous RG studies of Ref. [64] exploring the phase diagram of the antiferromagnetic version of Eq. (9), which also observed a significant decrease of $T_c$ with increasing ratio $J_2/J_1$.

Although an analysis of the full phase diagram of $J_2/|J_1|$ as a function of temperature is beyond the scope of this work, we emphasize that the results shown here do not represent an upper limit in terms of numerical feasibility. The typical MPS bond dimensions required to accurately simulate the width $N_y = 7$ systems are still small ($m < 120$) and handling the additional entanglement expected from increasing $J_2$ or decreasing $\Delta$ is definitely possible. In combination with the more efficient sampling routine of Ref. [55], the METTS-based scheme presented here offers a lot of potential to study thermal phase transition in frustrated lattice models without having to rely on perturbative approaches. Lastly, we note that calculations on the smaller clusters could have been carried out with density-matrix purification as well. But as METTS has much wider applicability to challenging systems, we refrained from using purification for this proof-of-principle study.

# 4    Conclusion

Despite much effort, the numerical treatment of strongly correlated electron systems in two dimensions remains a significant research challenge. Most methods to access finite-temperature physics of such systems are either limited to sign-problem-free models; cannot access very low temperatures; or fail in the vicinity of phase transitions. In this work we showed that tensor network techniques based on matrix product states are a compelling approach that can deal with all of these problems. And our approaches can be straightforwardly adapted to a very wide variety of systems beyond spin models and two-site interactions.

We employed a twofold strategy to deal with the system-size and temperature limitations of finite-temperature MPS techniques in the context of 2D systems. For high-temperatures we combined density-matrix purification with numerical linked-cluster expansions to get very accurate results for the thermodynamic limit. Since reaching low temperatures rapidly becomes prohibitive with purification, we applied the minimally entangled typical thermal state algorithm on cylinders to treat low-temperature regimes.

On a technical level, we elaborated on the treatment of finite-size effects in both finite-temperature MPS approaches. In this context, we note that NLCE could become an appealing companion to tensor network techniques on a more general level. By using a subset of possible clusters [28] and with recent progress in using NLCE for systems with long-ranged order [65], many other approaches such as ground-state DMRG and PEPS techniques could profit from the flexibility of the NLCE scheme. Another technical challenge we faced was the time evolution of 2D clusters within the MPS setup. We found that the combination of a Trotter decomposition with swap gates represents the best choice, currently, compared to a recently developed MPO scheme [30]. But MPO techniques will certainly continue to improve, and could quickly overtake the efficiency of the Trotter approach with the development of better algorithms, such as for applying an MPO to an MPS. Another approach could be to control costs or exploit additional parallelism by layering other types of Monte Carlo sampling on top of the METTS sampling; such an approach is explored for 2D systems in Ref. [66]. In the future it would be extremely helpful to MPS practitioners to establish whether there

exists a "best practice" approach among the various existing time evolution techniques for the treatment of long-ranged interactions.

As an application for our finite-temperature MPS techniques we treated the strongly frustrated spin-$\frac{1}{2}$ triangular Heisenberg antiferromagnet and established that our MPS techniques are competitive with other state of the art methods. We found that our MPS calculations are in excellent agreement with bold diagrammatic QMC and series expansions where results were available. We also studied the magnetic-field dependencies of different thermal properties. Furthermore, we showed that METTS is capable of treating finite-temperature phase transitions in the context of a frustrated spin-$\frac{1}{2}$ $J_1$-$J_2$ XXZ model on the square lattice. This approach shows potential for controlled calculations of the phase diagrams of a wide variety of frustrated lattice systems.

Going forward, we expect that tensor network techniques will play an important role in understanding the finite-temperature properties in many two-dimensional frustrated systems. Although our results are already very promising, we have only taken a first step in this direction. With continually improving algorithms to produce and sample METTS, we expect the METTS approach to become an ever more powerful for studying 2D systems. For example, a recent work provides a framework for exploiting symmetries when producing METTS for any Hamiltonian with conserved quantum numbers [55]. Also the ideas presented here are directly transferable to other tensor networks. For example, a finite-temperature PEPS construction [67, 68] could replace MPS as the cluster solver in NLCE. As a natural tensor network ansatz for 2D systems, PEPS should also be useful for extending the METTS technique to lower temperatures and larger systems.

*Acknowledgments* We thank Sergey Kulagin and Marcos Rigol, who were kind enough to provide reference data for the triangular lattice Heisenberg model. We thank Norm Tubman for providing us crucial computing resources to obtain the low-temperature triangular lattice $N_y = 6$ data. We also thank Juan Carrasquilla, Lode Pollet, and Matthias Punk for helpful discussions. B.B. thanks Andreas Weichselbaum and Jan von Delft for their support of this research, as well as the tensor network group at UCI for their hospitality at the initial stages of this work. This work used the Extreme Science and Engineering Discovery Environment (XSEDE) [69], which is supported by National Science Foundation grant number ACI-1548562. This research was supported by the DFG through the Excellence Cluster "Nanosystems Initiative Munich", SFB/TR 12, SFB 631 and by BaCaTeC Grant 15 [2014-2]. Z.Z., S.R.W., and E.M.S. were supported by the Simons Foundation Many-Electron Collaboration and by NSF grant DMR-1505406.

# A Numerical linked-cluster expansion

For completeness, we present a brief summary of numerical linked-cluster expansion (NLCE) and the modified embedding scheme that has been used to obtain the MPS-NLCE results in Sec. 3.1.

## A.1 Basics of NLCE

The key idea of NLCE is to obtain an extensive observable $O$ directly in the thermodynamic limit, while using measurements on finite-size clusters and eliminating boundary and finite-size effects by a systematic resummation strategy [25, 26, 27]. To this end, the expectation

value of $O$ per site can be represented by the sum of contributions of all different clusters, which can be embedded in the lattice $\mathcal{L}$,

$$O(\mathcal{L})/N = \sum_c L(c) \times W_O(c). \tag{11}$$

Each specific cluster $c$ contributes a certain weight $W_o(c)$ to the sum, which is multiplied by a combinatorial factor $L(c)$ defining the number of different ways to embed $c$ on the lattice. The weights are defined recursively by

$$W_O(c) = O(c) - \sum_{s \subset L(c)} M(s)W_O(s), \tag{12}$$

with $O(c)$ being the observable of interest calculated on cluster $c$. The sum runs over all subcluster $s$ that can be embedded into $c$ and the combinatorial factor $M(s)$ indicates in how many different ways this can be achieved. Eq. (12) can be interpreted as a generalization of the inclusion-exclusion principle and ensures that double counting of clusters is avoided [70].

To perform an NLCE calculation, one generates all relevant cluster starting with the smallest one without subclusters ($W_O(1) = O(1)$) up to some maximum size and evaluates $O(c)$ on each cluster. By truncating Eqs. (11) and (12) at the maximum cluster size, one obtains an approximation of the expectation value $O/N$ in the thermodynamic limit.

The quality of the result strongly depends on the correlation length in the system [70]. If the correlation length of the system is smaller than the maximum cluster size included, NLCE results show exponential convergence. This is reflected in the exponential decay of the weights $W(c)$ for clusters larger than the correlation length. Close to a phase transition or in an ordered phase at low temperatures, the correlations length typically exceeds the numerically accessible cluster size. In these cases, some properties show algebraic convergence (e.g. energy) while others (e.g. specific heat) might diverge and NLCE eventually breaks down. Even then, finite-size scaling or adapted summation techniques can help to extract useful information [26].

## A.2 Cluster groupings and order

Conventional NLCE calculations include all possible connected clusters up to a certain number of sites or bonds. Generating and embedding all relevant clusters and subclusters poses a numerical challenge. In fact, it can be shown that the cluster embedding problem relates to an NP-complete graph embedding problem. This numerical bottleneck limits conventional NLCE approaches for zero-temperature properties to $\sim 16$ sites.

However, NLCE can be formulated in multiple ways in terms of the clusters one chooses to include. It possible to converge Eq. (11) with an alternative cluster definition, as long as this is done in a self-consistent way. In other words, it should still be possible to decompose each cluster $c$ into subclusters $s$ according to Eq. (12), all subclusters being constructed according to the same alternative definition.

Based on this idea, Ref. [28] introduced an alternative cluster grouping for square lattice geometries based on rectangular clusters only. This modified grouping scheme is illustrated in Fig. 8 for a few examples and it drastically reduces the complexity of the cluster embedding problem. To illustrate this, consider all clusters up to a maximum of 16 sites on a square lattice system with NN interactions only. Including all connected clusters, one ends up with

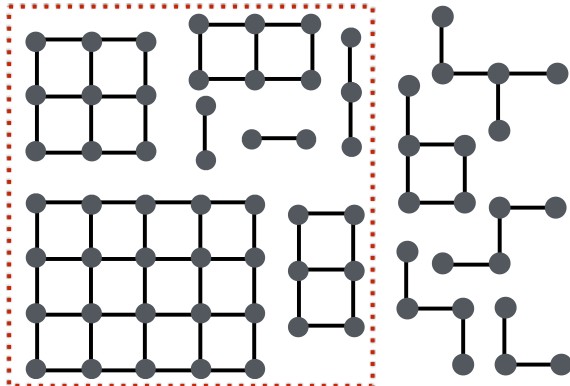

Figure 8: Example of clusters entering a NLCE calculation on the square lattice. The cluster grouping scheme of Ref. [28] only considers rectangular clusters (encircled red), significantly reducing the complexity of the cluster embedding problem.

$\mathcal{O}(10^8)$ clusters, whereas the restriction to rectangles reduces this number to a total of 27 clusters. Thus, the rectangular cluster grouping shifts the numerical bottleneck entirely to the cluster solver. Now the maximum expansion order of the NLCE is only limited by the size of cluster on which the observable $O(c)$ can still be measured. Employing this grouping scheme at zero temperature, Ref. [70] reached system sizes of $\sim 50$ sites based on Lanczos and DMRG cluster solvers in their NLCE calculations.

As discussed in the main part of this work, we combine NLCE with density-matrix purification as cluster solver and extend finite-temperature calculations previously performed by exact diagonalization. Following Refs. [28] and [70], we employ a rectangular cluster grouping in all calculations and perform the NLCE with a quadratic ordering scheme. This means that we include all clusters fitting inside the largest quadratic cluster considered, further reducing the number of clusters entering the calculation.

### A.3   Failure of statistical cluster solvers

Measurements of an external quantity $O$ performed with statistical approaches such as METTS or QMC include a statistical error scaling as $\delta O \sim \sqrt{\mathrm{Var}[O]}/\sqrt{M}$ where $M$ is the sample size. One is typically interested in the value of $O$ in the thermodynamic limit but measurements have to be performed on finite-size systems with $N$ sites with subsequent extrapolation $N \to \infty$. Standard finite-size scaling divides the value of the observable by $N$ and extrapolates $O/N$ as a function of system sizes. In this procedure the absolute value of the statistical error is obviously also reduced by the system size. In other words, a relative error $\delta O/O$ in the bare measurement of $O$ on a cluster remains the same when computing $O/N$ - a trivial statement.

To compute $O/N$ in the framework of NLCE, the bare measurement $O(c)$ on each cluster $c$ (not $O(c)/N(c)$!) enters the series in Eq. (11) multiple times. Hence, statistical fluctuations which may seem small compared to the bare value of $O(c)$ on a large cluster can become much more pronounced since the absolute error $\delta O(c)$ might not be small compared to $O/N$ in the thermodynamic limit. This fact renders any statistical cluster solver for NLCE inapplicable.

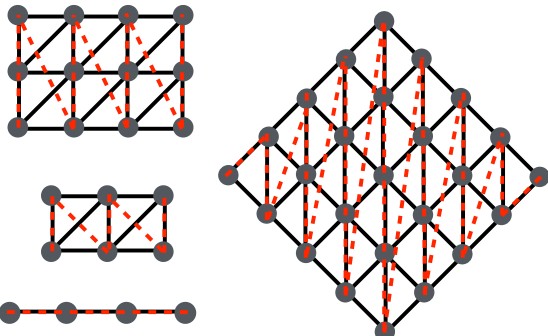

Figure 9: Example of clusters entering our rectangular-based NLCE scheme for the triangular Heisenberg lattice. The dashed red lines indicate the quasi-one-dimensional MPS path through the clusters. Rotating square clusters by 45° enables us to use an adapted MPS path that reduces the overall entanglement of the resulting MPS.

## B    Numerical details

Below we summarize additional numerical details of the MPS results presented in sections 3.1 and 3.2. All calculations are performed using the ITensor library [37].

### B.1    Triangular lattice Heisenberg model

*MPS-NLCE.–* We employ standard density-matrix purification as finite-temperature cluster solver for the MPS-NLCE on the triangular lattice. Specifically, we rely on a second order Suzuki-Trotter decomposition to carry out the imaginary-time evolution with time step $\tau = 0.1$ and the truncation cut-off $\epsilon = 10^{-10}$. This results in a maximum bond dimension $m < 8500$ for $\beta = 1.4$ on largest cluster considered in our calculations (size $5 \times 5$). We always exploit U(1)-spin symmetry in the purified setup and use open boundary conditions in both directions.

To set up the NLCE with density-matrix purification on the triangular lattice, we employ the rectangular cluster grouping and perform the NLCE with a quadratic ordering scheme, as specified in Sec. A.2. A few examples of rectangular clusters entering the calculation are displayed in Fig. 9. The dashed red lines in Fig. 9 illustrate the choices of MPS paths through the cluster. We use an adapted path on square clusters, rotating them by 45°, to reduce the costs associated with representing an entangled 2D state as an MPS.

*METTS.–* For our METTS results we employ a second order Suzuki-Trotter decomposition with $\tau = 0.1$ and for the imaginary-time evolution set the truncation cut-off $\epsilon = 10^{-10}$ for width $N_y = 3$ and $N_y = 4$ systems and $\epsilon = 10^{-9}$ for $N_y = 6$ systems. All measurements are computed independently on length $N_x = 8$ and length $N_x = 16$ cylinders, before a bulk-cylinder extrapolations is used to minimize finite-length effects [29]. The METTS collapse into a product state is performed with a maximally mixed basis set [18]. In addition, for $h_z = 0$ we exploit the SU(2) spin rotation symmetry to implicitly rotate the basis states back into the $\hat{S}^z$ basis after each collapse, allowing us to use a more efficient total-$S^z$ conserving block-sparse representation for the time evolution. METTS sample sizes $M$ vary between $M \approx 500$ for $T = 0.25$ on the width $N_y = 4$ cylinders to several thousand METTS for higher temperatures. We refrained from extrapolating the data as a function of the cylinder width

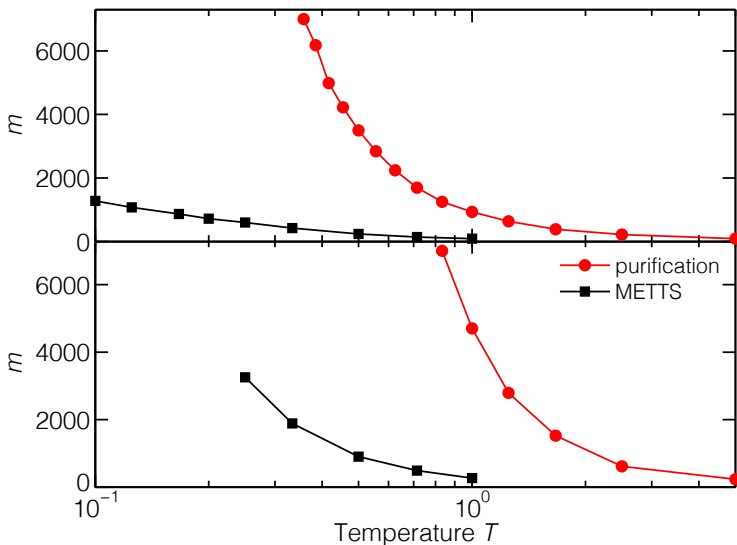

Figure 10: Maximum bond dimension $m$ required by METTS (black) and density-matrix purification (red) for simulating the triangular AFHM on a cylinders of sizes (a) $16 \times 3$ and (b) $16 \times 4$ at various temperatures and zero magnetic field, using the numerical parameters specific in the text.

since the high amount of entanglement in the model limits us to width $N_y = 6$ systems.

To give some perspective on the numerical feasibility of the METTS calculations for the specified parameters, Fig. 10 displays the maximum bond dimension $m$ required by METTS and density-matrix purification for two cylinders of sizes $16 \times 3$ and $16 \times 4$ at various temperatures and zero magnetic field. The METTS samples at high temperatures can be represented by MPS with small bond dimensions $m < 1000$, only the low-temperature samples on the width $N_y = 4$ cylinders require $m \geqslant 2000$ resulting in significant numerical effort (CPU times of several hours on four cores in order to generate single METTS). In these cases we made significant use of parallelizing the METTS sampling on numerous machines. With density-matrix purification applied to the same system, one is limited to temperatures $T_{\min}^{\mathrm{Pur}} > 0.35$ on the width $N_y = 3$ and $T_{\min}^{\mathrm{Pur}} > 0.8$ on the width $N_y = 4$ cylinders, respectively. For lower temperatures, we have to retain an unfeasibly large $m \gg 7000$ to keep the truncation error in the time evolution constant. The accessible temperature ranges for purification have been included as guide for the eye in Fig. 2 and illustrate the importance of the METTS algorithm to cover the low-temperature regime, where purification is no longer feasible.

### B.2   $J_1$-$J_2$ XXZ model on square lattice

All METTS calculations for the $J_1$-$J_2$ XXZ model on the square lattice are performed with $\tau = 0.1$ and $\epsilon = 10^{-10}$. We typically have to obtain large sample sizes $M > 10000$ to converge the statistical error of the fourth order moment entering the Binder cumulant formula Eq. (10). Moreover, we perform measurements only in the middle half of the system and add a local pinning field $-\frac{\Delta}{2}\hat{S}_i^z$ on both ends of the cylinder to select one order parameter direction. The latter is of great importance, as calculations without pinning field can favor domain-wall formation on the cylinder that leads to ambiguous results and preclude the precise

determination of $T_c$.

## B.3   Statistical error bars

The statistical error bars of the METTS calculations shown in Figs. 2, 4, 5, and 7 are derived using the standard error for energy density and magnetization. For more complex observables, such as specific heat, susceptibility and the Binder cumulant, we employ a resampling procedure via the bootstrap method. In this way, we properly take into account correlations between first, second, and forth moments of a bulk observable [18]. Standard Gaussian error addition is employed in the context of the infinite-cylinder extrapolation.

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
