# Peer review of "Matrix product state techniques for two-dimensional systems at finite temperature"

_SciPost Physics_

## Round 2 · Referee Report · Anonymous (Referee 1) · 2017-9-22

Strengths

1) The paper demonstrates the applicability of an unbiased numerical method (DMRG) for 2D quantum many body systems at finite temperatures 2) Results obtained for a frustrated 2D spin model based on the Binder cumulant demonstrate high precision in determining phase transition points 3) Demonstration of capability to study finite temperature properties of paradigmatic S=1/2 AFM Heisenberg model on triangular lattice

Weaknesses

1) Confusing presentation/discussion of both numerical approach and results 2) Poor error discussion for the comparison between the used time-evolution schemes and in particular misleading conclusions about their accuracy 3) Presented data for paradigmatic S=1/2 AFM Heisenberg model on triangular lattice lacks discussion

Report

DMRG has demonstrated its power in performing both zero- and finite-temperature calculations in 1D systems in the past 25 years in a variety of applications. However, inhibited by the structure of the underlying tensor network, extending the method to 2D systems is way more difficult, posing the question of algorithmic schemes which may eventually push DMRG to become a powerful tool in higher dimensions too. The manuscript addresses this question by combining cluster techniques (NLCE) and a statistical sampling method (METTS) with imaginary time evolution in the framework of matrix product states to extract finite temperature properties of two dimensional systems in the thermodynamic limit. The authors also present a comparison of a recently proposed time evolution scheme by Pollmann et al. [PRB 91, 165112 (2015)] to the Suzuki-Trotter decoupling with swap gate application. Both schemes are applied to the (S=1/2) antiferromagnetic Heisenberg model on the triangular lattice and the (J_{1}-J_{2}) XXZ model on a square lattice.

Given the complexity of the problem, seeing methodological progress is very welcome and the manuscript should be published. However, in its present form it contains a large number of weaknesses so that it needs substantial revision prior to publication, as detailed in the following.

The manuscript presents a suitable application of current DMRG techniques to 2-dimensional problems providing evidence that a combination of stochastical and cluster methods with DMRG can yield competitive algorithms to obtain unbiased finite temperature results on frustrated systems in the thermodynamic limit. However, presentation can be improved in several points:

  • From the comparison of the MPO representation of the time evolution operator [PRB 91, 165112 (2015)] with the Suzuki-Trotter decoupling with swap gate application the conclusion is drawn that "Suzuki-Trotter decomposition with swap gates is currently the most accurate and efficient". This is a strong statement and misleading: In [PRB 91, 165112 (2015)] two MPO representations labeled (W_{I}) and (W_{II}) are used, of which ( W_{I} ) is argued not to be as precise and well-suited for time evolution as (W_{II}) (see the section there between eqn. 8 and 9). In this manuscript, however, only (W_{I}) is used. Since this is the weaker of the two schemes, this prohibits drawing the general conclusion, that the Suzuki-Trotter method is favorable.

  • The approach used to do the comparison of the two methods is questionable: The accuracy of both time evolution methods is compared by computing the overlap of the time evolved state with respect to a "quasi-exact reference state", which, however, is also obtained via Suzuki-Trotter decoupling, but with smaller time step. This is in mismatch with the discussion in the main text (see figure 1a), where the authors argue that the accuracy of Suzuki-Trotter calculations is decreased by reducing the time step due to the larger number of performed truncation steps. Having this in mind, it remains questionable, if the calculation with smaller time step should be used as reference at all.

  • It remains unclear, why the overlap of two MPS obtained with completely different approaches should be a meaningful measure of error at all: on general grounds, it is to be expected that two MPS obtained with the same scheme, but different computational parameters, are closer to each other (i.e., larger overlap), than two MPS obtained with completely different schemes (e.g., the updates are obtained using different optimization algorithms, the SVDs result in different transformation matrices etc.).

  • The comparison shown in Fig. 1 would be more meaningful if comparison to exact diagonalization results for observables would be shown (for the XX-model one should be able to treat large system sizes). This would probably lead to a more informative error estimate for the different approaches used.

  • From the text it remains unclear how the swap gates are actually applied, and how the accumulated deviation from the target state is estimated (the given references 12,18 discuss either PEPS or METTS algorithms and not the present case).

  • At various points the presentation can be further improved:

  • p.6: For 2D clusters long-ranged interactions are requiring contractions of several swap gates. How can this conserve the accuracy of the pure nearest neighbour Trotter-Suzuki in finite precision arithmetics?
  • p.8: The MPO (W_{I}) is not capable to treat powers of local operators on 1 site (e.g. (S^{z}^{(k)} )), which can be inferred from PRB 91, 165112 (2015) (section between eqns. 8 and 9). This should be discussed on page 8 when the properties of W_{I} are explained.
  • p.10: Fig.2b a solid ocher line is shown but not mentioned in any of the legends, does it refer to the HTSE-dashed dotted line?
  • p.11: Cylinder calculations are mentioned to be done for cylinders of y-size (N_{y}=3,4,6 ), why not also for (N_{y}=5)?
  • p.17: The computational effort and sample variance as function of the temperature is discussed, but no graph is shown. It would be helpful for the reader to provide an example plot.

As a general remark, one has a hard time to follow the presentation of the results in sec. 3. For example, it is confusing that the legend in fig.2 is split over three subplots.

Furthermore, figs. 2a and 2c are containing METTS data for cylinder sizes (N_{y} = 3,4,6 ), while fig. 2b misses (N_{y}=6). This is confusing: the trend of the METTS data for (N_{y}=3,4 ) below (T<0.7) misses the (assumed) HTSE with a deviation, which is larger than the given error bounds. In fact, for some temperature region the data for (N_{y}=3) is strictly above the (assumed) HTSE data while the points corresponding to (N_{y}=4) are strictly below. If this is a finite size effect, the question arises wether (N_{y}=6 ) results might be even below the (N_{y}=4) data? Following the description of the simulation parameters one cannot find the value of ( N_{x}) concerning the cylindrical geometry. Also, on p.13, only data for (N_{y} = 4 ) is shown. It would be informative to show results for the further sizes discussed previously, (N_{y} = 3,6 ).

In figs. 4c,d the MPS-NLCE seems to collapse earlier for the computationally more involved quantities. What kind of calculation strategies were employed to obtain the expectation values of ( C_{v},\chi_{z} )? Perhaps more elaborate numerical schemes as discussed in [Phys. Rev. B 95, 035129 (2017)] or arXiv:1706.05338 might help in improving the accuracy/range of applicability of this approach.

Requested changes

1) Improve discussion of the results as detailed in the report (e.g., Fig. 2,...) 2) Reconsider or better justify the statement, that the Trotter-approach together with swap-gates is the most accurate and efficient. 3) Improve the comparison of the time evolution schemes to 'exact' results by providing a comparison of observables and the overlap obtained by exact (ED) approaches

---

## Round 2 · Referee Report · Anonymous (Referee 2) · 2017-9-29

Strengths

1) Very detailed and complete benchmark study of different MPS techniques for finite temperature. 2) Benchmark of different imaginary time evolution techniques for long range interactions.

Weaknesses

None

Report

The authors present a very detailed and extensive study of two major techniques
for the simulation of two dimensional quantum systems at finite temperatures.
They review both minimally entangled typical thermal states (METTS) and density
matrix purification and compare their performance in detail, demonstrating that
METTS is viable at much lower temperatures than can be reached by purification.
They also discuss the application of density matrix purification as a cluster
solver for the numerical linked cluster expansion (NLCE), limited to rectangular
clusters in order to avoid the NP hard cluster embedding problem.

The paper is very well written and pedagogical, providing valuable benchmark
results in order to determine which method is best suited for specific problems.
In particular, it is found that for the application as an NLCE cluster solver,
purification is superior to METTS due to the absence of statistical errors,
while METTS can reach much lower temperatures with fixed bond dimensions.

The application of MPS techniques to two dimensional problems naturally
introduces an effective long range Hamiltonian. Two techniques are compared: A
combination of a Trotter decomposition of the (imaginary) time evolution
operator and swap gates and a novel MPO representation of the time evolution
operator. For the tested cases and system sizes, the Trotter scheme was found
superior and therefore used to benchmark the finite temperature methods for two
different models: The triangular lattice Heisenberg model and the J1-J2 model on
the square lattice. For the first (frustrated) model, METTS and MPS-NLCE
(purification) results are compared to state of the art results from Bold line
diagrammatic QMC as well as High temperature series expansions, yielding
excellent agreement. For the J1-J2 model, the finite temperature phase
transition to a low temperature ferromagnetic phase was analyzed carefully,
yielding excellent agreement with QMC results for the critical temperature in
the non-frustrated case using a Binder cumulant.

Overall, I find this work very interesting, timely, scientifically sound and
complete and recommend it for publication in SciPost Physics. I list minor
comments below, which the authors may want to consider.

Requested changes

1) For completeness, in the list of references (2nd paragraph of introduction) for special cases where QMC was shown to be viable even in frustrated systems, a reference to [Phys. Rev. B 93, 054408 (2016)] and [SciPost Phys. 3, 005 (2017)] could be added.

2) In the discussion of methods, the background could be made more explicit: why is the maximally entangled state between the physical space and the ancilla space reproducing the infinite temperature density matrix (proportional to identity)? A quick explanation how the propagation of this state in imaginary time on the physical degrees of freedom reproduces exp(-beta H) after tracing out the ancilla would also help the novice understand the concept.

3) In Figs 5 and 7, the extracted value of T_c could be shown in all panels.

4) How is the crossing of the Binder cumulant analyzed?

5) In Fig 10, the labels a) and b) are missing.

---

## Editorial Decision

awaiting_resubmission